# Salivary Biomarkers and Their Correlation with Pain and Stress in Patients with Burning Mouth Syndrome

**DOI:** 10.3390/jcm9040929

**Published:** 2020-03-28

**Authors:** Pia Lopez-Jornet, Candela Castillo Felipe, Luis Pardo-Marin, Jose J. Ceron, Eduardo Pons-Fuster, Asta Tvarijonaviciute

**Affiliations:** 1Department Stomatology School of Medicine, Biomedical Research Institute (IMIB-Arrixaca), Faculty of Medicine and Odontology, University of Murcia, 30008 Adv Marques de los Velez s/n, Spain; 2Department Stomatology School of Medicine, Faculty of Medicine and Odontology, University of Murcia; 30008 Murcia, Span; candela.casti@gmail.com; 3Interdisciplinary Laboratory of Clinical Analysis, Interlab-UMU, Regional Campus of International Excellence ‘Campus Mare Nostrum’, University of Murcia, Espinardo, 30100 Murcia, Spain; lpm1@um.es (L.P.-M.); jjceron@um.es (J.J.C.); asta@um.es (A.T.); 4Biomedical Research Institute (IMIB-Arrixaca), Faculty of Medicine and Odontology, University of Murcia, 30008 Murcia, Span

**Keywords:** saliva, IgA, alpha-amylase, uric acid, stress, inflammation

## Abstract

Objective: To evaluate a panel of salivary analytes involving biomarkers of inflammation, stress, immune system and antioxidant status in patients with burning mouth syndrome (BMS) and to study their relationship with clinical variables. Materials and Methods: A total of 51 patients with BMS and 31 controls were consecutively enrolled in the study, with the recording of oral habits, the severity of pain using a visual analogue scale (VAS), the Hospital Anxiety and Depression (HAD) score and the Oral Health Impact Profile-14 (OHIP14) score. Resting whole saliva was collected with the drainage technique, followed by the measurement of 11 biomarkers. Results: The salivary flow was higher in patients with BMS. Among all the biomarkers studied, significantly higher levels of alpha-amylase, immunoglobulin A (IgA), and macrophage inflammatory protein-4 (MIP4) and lower levels of uric acid and ferric reducing activity of plasma (FRAP) were observed in the saliva of patients with BMS as compared to the controls (*p* < 0.05 in all cases). Positive correlations were found between pain, oral quality of life and anxiety scores and salivary biomarkers. Conclusions: BMS is associated with changes in salivary biomarkers of inflammation, oxidative stress and stress, being related to the degree of pain and anxiety.

## 1. Introduction

The International Association for the Study of Pain (IASP) defines burning mouth syndrome (BMS) as an intraoral burning or dysesthetic sensation that manifests daily for more than two hours over three months, with no evidence of clinical lesions [1]. The epidemiological data on BMS are largely contradictory, in part because of a lack of strict compliance with the diagnostic criteria of the disorder. Nevertheless, BMS is estimated to affect 4% of the general population and 18–33% of all postmenopausal women [2,3,4,5,6]. The symptoms are generally focused on the tongue and lips and are almost always bilateral. The palate and other locations within the oral cavity are less commonly affected. Although much has been published about BMS, its underlying etiopathogenesis is largely unknown, and complex interactions among local, systemic and psychogenic factors are believed to be involved [2,7,8]. For these reasons, BMS is poorly diagnosed, and in many cases management is deficient, a situation that causes frustration for both physicians and patients [3,5]. In the same way as with other chronic pain syndromes, BMS is characterized by associated psychological problems. Changes in personality and mood state, particularly anxiety and depression, are a typical finding in patients with BMS [2,9].

The use of saliva as a sample for clinical analysis has increased in recent decades. Saliva is a body fluid that reflects the physiological condition of the body. It can be easily collected using non-invasive and relatively inexpensive methods [4,5], and it is gaining attention as a source of biomarkers in different oral and systemic pathologies [10]. In BMS an increase in amylase (a marker of the adrenergic system) and immunoglobulin A (IgA) (a marker of the immune system) have been found [11,12,13,14]. However, to the authors’ knowledge, there have been no studies in which a panel of analytes reflecting possible pathogenic and psychological factors were evaluated in this disease, and also compared with the degree of stress and pain.

The objective of this study was to evaluate how a panel of biomarkers of inflammation represented by complement C4 (CC4), α1-antitrypsin (a1AT), C-reactive protein (CRP), macrophage inflammatory protein-4 (MIP4), pigment epithelium-derived factor (PEDF), serum amyloid P (SAP), haptoglobin (Hp), a panel of biomarkers of oxidative stress integrated by uric acid and ferric reducing activity of plasma (FRAP), the salivary alpha-amylase (sAA) as a marker of the adrenergic system and total immunoglobulin A (IgA) as a marker of the innate immune system can be altered in patients with BMS, and study their possible relation with the degree of pain and stress of the patients was measured by visual analogue scale (VAS) and the Hospital Anxiety and Depression (HAD) score. Also, the possible influence of oral health was evaluated.

## 2. Material and Methods 

### 2.1. Study Design and Subjects

A cross-sectional study was carried out with data compiled between January 2018 and September 2019 in the Dental Clinic of the University of Murcia (Murcia, Spain). All the patients were informed about the study and gave consent to participation in the trial, which was approved by the local Clinical Research Ethics Committee (Ref.: 2203/2018). The following inclusion criteria were established: patients over 18 years of age with a diagnosis of BMS based on exhaustive oral examination, laboratory findings and the patient profile and symptoms [15], presenting continuous symptoms of oral burning or pain persisting for at least two hours per day, lasting longer than three months, without paroxysms and not following any unilateral nerve trajectory, and with no clinical mucosal alterations. The patients underwent blood testing to confirm the absence of alterations (including blood count, glucose, serum iron, ferritin and transferrin, folic acid and vitamin B12 levels). Cancer patients and individuals with known liver or kidney disease were excluded, as were those with oral diseases other than BMS such as Sjögren’s syndrome or ongoing infection, thyroid diseases, coagulation disorders, psychiatric disease, and pregnant or nursing women. 

The healthy controls were recruited from the Dental School of the University of Murcia and consisted of patients with the same sociosanitary characteristics and matched for age and gender. Following the procurement of written informed consent, a structured questionnaire was administered to confirm the absence of significant medical conditions. The study was carried out following the STROBE (Strengthening the Reporting of Observational studies in Epidemiology) guidelines.

In all cases, patients were not included in the study if their samples showed hemolysis as determined by visual inspection [16,17].

### 2.2. Data Collection

A structured interview was used to collect sociodemographic and clinical information as well as data referring to smoking and alcohol consumption. The patients were evaluated by a trained professional (C.C.F.) who conducted the interviews and explorations, administered the questionnaires and collected the saliva samples.

An extraoral and intraoral exploration was carried out, documenting the presence of caries and missing teeth. Pain intensity was scored using a visual analogue scale (VAS) (0 = no pain and 10 = worst possible pain) [18].

The Hospital Anxiety and Depression (HAD) scale [19] was used to assess the emotional state of the participants. This instrument consists of two subscales relating to anxiety (HAD-A) and depression (HAD-D). Concerning the interpretation of the HAD scale, scores of > 10 indicate the probable presence of anxiety or depression, scores of ≤ 7 indicate no significant anxiety or depression, and scores of 8–10 are of borderline significance.

The evaluation of the oral quality of life, in turn, was based on the Oral Health Impact Profile-14 (OHIP14) score [20].

### 2.3. Saliva Collection

Resting whole saliva was collected using a standardized method [18]. To avoid possible contamination from other sources, the patients were instructed to rinse the mouth thoroughly before saliva sample collection. The subjects were required to avoid heavy physical exercise one hour before sampling. Unstimulated saliva was obtained using the draining method for 5 min. The samples were collected at about the same time in all subjects (8:00 to 11:00 a.m.). The saliva was vortexed and centrifuged (3000× *g* for 10 min at 4 °C) immediately after collection, and the supernatant was transferred into polypropylene tubes and stored at −80 °C until analysis.

### 2.4. Biochemical Analysis

Complement C4, a1AT, CRP, MIP4, PEDF and SAP in saliva were analyzed using a commercially available kit (Human Neurodegenerative Disease Magnetic Bead Panel 2, Neuroscience Multiplex Assay; Life Science, Darmstadt, Germany) according to the manufacturer’s instructions. Values were calculated based on a standard curve constructed for the assay. Saliva total protein quantification was done using a commercially available colorimetric kit for measuring urine and low-complexity region (LCR) proteins (protein in urine and CSF, Spinreact, Barcelona, Spain) and validated for human saliva [21]. Uric acid was measured using a colorimetric commercial kit (Uric acid, Beckman Coulter Inc., Fullerton, CA, USA) following the International Federation of Clinical Chemistry and Laboratory Medicine (IFCC) method [21]. Total IgA was evaluated with a commercial ELISA kit (Bethyl, Montgomery, TX, USA) previously validated for use with human saliva samples [18]. FRAP (ferric reducing ability of plasma) measurement was based on the method described by Benzie and Strain [22] with some modifications [23]. Hp levels were measured in saliva using a homemade immunoassay as previously described [24]. Salivary alpha-amylase (sAA) activity was measured using a colorimetric commercial kit (Alpha-Amylase, Beckman Coulter Inc., Fullerton, CA, USA) following the IFCC method as previously reported and validated [25].

### 2.5. Statistical Analysis

For the descriptive statistical analysis of the sample, the basic descriptive methods were used, with calculation of the frequencies, mean, median, standard deviation (SD) and 25th and 75th percentile values. The comparison of means between groups was based on the Student’s t-test or Mann–Whitney test, depending on the data distribution as verified with the Kolmogorov–Smirnov test, and the homogeneity of groups was confirmed. Clinical scores, smoking and alcohol consumption habits, as well as sex distribution among the two groups, were explored using the chi-square test. The correlations between variables were checked using the partial correlation corrected by age and sex. Previous studies that also analyzed biomarkers in saliva in patients with burning mouth syndrome, such as sAA and total IgA, have been able to detect differences with 30 or fewer individuals in each group considering α = 0.05 and β = 0.20 [11]. Based on these data we assumed that the study was sufficiently powered (n_controls_ = 31; n_BMS_ = 51) to achieve our aims. Differences and associations among groups were considered statistically significant when *p* < 0.05.

## 3. Results

The study sample consisted of 82 consecutively enrolled individuals (71 females and 11 males), of which 51 were diagnosed with BMS while the remaining 31 constituted the control group. Table 1 describes the characteristics of each group. Statistically significant differences were not detected in sex, age or smoking and alcohol consumption habits between the two groups of patients (*p* > 0.05).

Concerning oral hygiene and health, the controls and patients with BMS did not present statistically significant differences in habits related to tooth brushing and use of mouthwashes, while patients with BMS used more dental floss than controls (*p* < 0.01). In turn, controls presented a comparatively greater number of caries, though the difference was not statistically significant (*p* > 0.05), and significantly fewer missing teeth compared with the BMS group (*p* < 0.01).

Of the clinical variables, the mean burning sensation score in patients with BMS was greater than 8, while it was equal to 0 in healthy controls. The OHIP14 score was 2.3-fold higher in the BMS group as compared with the control group (*p* = 0.001).

Concerning the psychological profile as evaluated by the HAD scale, the anxiety scores among the patients were 2.3-fold higher than in controls (*p* < 0.001). While HAD-D, although being 1.4-fold higher in patients, did not show statistically significant differences between the two groups (*p* > 0.05).

Resting whole saliva flow was 1.4-fold greater in the BMS group as compared to healthy controls (*p* < 0.05) (Figure 1a and Appendix A). The study of biomarkers in resting whole saliva yielded significant differences between the cases (patients with BMS) and controls for sAA (*p* < 0.01) and IgA (*p* < 0.05) (Figure 1a,b and Appendix A).

When absolute values of salivary biomarkers were corrected by salivary flow, sAA, IgA and MIP4 showed statistically higher concentrations in patients with BMS as compared with healthy controls (Figure 2 and Appendix A). When values were corrected by total protein content, statistically significantly higher levels in patients with BMS were observed for sAA and lower for uric acid and FRAP (Figure 3 and Appendix A).

Partial correlation data are given in Appendix A. When absolute values were evaluated, a positive correlation was detected between VAS and total proteins (*r* = 0.241; *p* < 0.05), sAA (*r* = 0.260; *p* < 0.05) and IgA (*r* = 0.334; *p* < 0.01); HAD-A was correlated with total proteins (*r* = 0.251; *p* < 0.05), IgA (*r* = 0.302; *p* < 0.05), PEDF (*r* = 0.351; *p* < 0.01) and MIP4 (*r* = 0.267; *p* < 0.05) and HAD-D with IgA (*r* = 0.261; *p* < 0.05), PEDF (*r* = 0.363; *p* < 0.01) and MIP4 (*r* = 0.271; *p* < 0.05); OHIP14 was correlated with IgA (*r* = 0.273; *p* < 0.05).

After data were corrected by salivary flow, statistically significant correlations remained between VAS and sAA (*r* = 0.269; *p* < 0.05) and IgA (*r* = 0.367; *p* < 0.01); between HAD-A and IgA (*r* = 0.338; *p* < 0.01), PEDF (*r* = 0.297; *p* < 0.05) and MIP4 (*r* = 0.264; *p* < 0.05); between HAD-D and PEDF (*r* = 0.386; *p* = 0.01) and MIP4 (*r* = 0.269; *p* < 0.05) and between OHIP14 and IgA (*r* = 0.313; *p* < 0.01).

After data were corrected by salivary flow, negative correlations appeared between sAA and OHIP14 (*r* = −0.242; *p* < 0.05) and between HAD-A and FRAP (*r* = 0.250; *p* < 0.05) and CRP (*r* = −0.243; *p* < 0.05).

## 4. Discussion

Burning mouth syndrome is a clinically relevant form of chronic orofacial pain [1,2,3]. The diagnosis of the syndrome remains a challenge for health professionals due to the discrepancy between the intensity of pain as reported by the patient and the absence of objective clinical lesions. In this study we therefore focused on the identification of potential biomarkers capable of reflecting the physiopathological changes involving pain, psychological stress and inflammation that occur in the context of the disease. Of 11 biomarkers evaluated in the present study, three—sAA, IgA and MIP4—were found to be increased, and two—uric acid and FRAP—were found to be decreased in the saliva of patients with BMS versus controls, suggesting that these patients can present with alterations in immune response, pro-inflammatory status and oxidative stress that are related to pain sensation and psychological stress.

In the present study, a higher salivary flow rate was detected in patients with BMS as compared with healthy controls. In the scientific literature, controversy exists regarding this topic since some authors did not detect changes in salivary flow between controls and patients with BMS [12,13,14], while others observed lower flow rates in patients with BMS [9,11,26]. Similarly, different authors reported divergent results related to total protein content in the saliva of patients with BMS, as some of them detected decreased [27], increased [12,14] or unchanged (present study) salivary total protein content in these patients when compared with controls. The characteristics of the method or device used to collect the samples and the analytical assays used may have influenced the results obtained and, therefore, the differences among the different studies. It is therefore important to follow the same guidelines for sample collection and processing to enable comparison of the results from different studies.

Since components present in saliva have different origins, i.e., are locally secreted by salivary glands or oral mucosa, or pass from blood among other, and can be affected by non-uniform salivary flow rate, the need to correct values of salivary biomarker by flow rate, total protein content or salivary osmolarity versus use of their absolute value is being increasingly acknowledged and studied by different authors [25,28]. In this study we aimed to evaluate the results without any adjustment, and we also corrected salivary flow or total protein content to gain knowledge about the possible effect that these corrections can have in this particular disease. When levels of biomarkers without any adjustment by saliva flow or protein content were compared between healthy and diseased patients, statistically significant changes were detected only in sAA and IgA. Correction by salivary flow rate allowed identification of the increased levels of MIP4 in the saliva of patients with BMS but also yielded higher differences between the studied groups for IgA in accordance with previously reported data [29]. When values were corrected by total protein content the presence of oxidative stress in patients with BMS was detected, as levels of uric acid and FRAP were lower compared to controls. Therefore, in BMS, correcting salivary values by flow rate and total protein content would be recommended in this disease since it can detect additional changes in salivary analytes that are correlated with the clinical condition of the patient.

sAA is considered to be a sensitive biomarker of stress-related changes that reflect the activity of the sympathetic nervous system (SNS) [30]. Most patients with BMS experience chronic pain and have a poorer quality of life than do healthy individuals [9]. Furthermore, chronic pain characterizing BMS was associated with psychological problems, specifically personality and mood changes, anxiety and depression. The results of this study agree with these findings and previous reports that detected higher sAA levels in the saliva of patients with BMS [12,14] since patients included in the present study showed both higher anxiety scores and sAA levels; however, another study did not detect changes in sAA between healthy controls and patients with BMS, and differences in the severity of the disease in the population evaluated or in the assay used could be the cause of these differences [9]. A weak positive correlation between sAA (expressed in absolute concentrations and when corrected by flow) and burning sensation was detected, which could indicate that increases in this enzyme could be influenced in part by pain.

Salivary IgA was another biomarker that showed statistically significant differences among controls and patients in the studied population, being higher in patients with BMS. These data are consistent with those reported by other authors [12,13,14]. Furthermore, a higher difference between the two groups was detected when IgA was corrected by salivary flow resulting in an increase of 1.8-fold to 2.9-fold change between the two groups. IgA is an immune glycoprotein that acts as a defense against pathogens [31]. Nevertheless, different studies have suggested that changes in salivary IgA can also be associated with stress [32]. This hypothesis would be supported by the observed positive correlations between IgA in saliva and oral quality of life, burning sensation and anxiety in the studied population. In addition, MIP, a biomarker of the innate immune system [33], was also increased in patients with BMS and weakly positively correlated with anxiety and depression scores, suggesting that the immune system is implicated in the pathogenesis of BMS.

In the present study, when the results were corrected by protein content, patients with BMS had lower salivary FRAP and uric acid, suggesting the presence of oxidative stress, and in the case of FRAP, it was weakly correlated with HAD-A, which would indicate that this increase could be influenced by stress. The two markers are closely related since uric acid was the main component (up to 60%) of the FRAP assay [22]. Uric acid is considered to be the most important antioxidant molecule in saliva since it is responsible for approximately 70% of the total antioxidant capacity of this body fluid. As a reactive oxygen species scavenger, it helps stabilize arterial pressure and oxidative stress [21,34,35,36]. In turn, alterations in its levels in saliva were associated with acute stress [34] and different local and systemic pathologies such as oral lichen planus [37] or nephropathies [38]. Previous studies that evaluated salivary uric acid in patients with BMS did not find statistically significant changes as compared with healthy controls [16,20], although these studies did not make corrections for total protein content.

A limitation of the study is the fact that the measurements were limited to a single time-point. Variations of these markers in the same individual should be investigated in the context of future studies with a larger sample size to increase the power, and by employing multiple samplings over time, resulting in more accurate estimations. Furthermore, it would be of interest to apply the biomarkers that showed significant changes in a larger population to corroborate the results of this study. Finally, the possible blood contamination in saliva samples should have been evaluated using more sensitive methods, such as determination of hemoglobin or transferrin, although these methods were shown to be affected by different factors such as age, hormones and salivary flow among others [16]. Conversely, visual inspection was shown to be sufficient in the case of determining some of the analytes including oxidative stress markers in saliva without interfering with the results [17].

## 5. Conclusions

In conclusion, patients with BMS showed changes in biomarkers associated with stress such as sAA and IgA, with the immune system such as MIP4 and oxidative stress such as uric acid and FRAP in saliva as compared with healthy controls, which are related to clinical variables including burning sensation and anxiety. Moreover, in this particular disease, the study of the absolute values but also the values corrected by flow and total protein would be recommended. Overall, biomarkers that were shown to change differently between groups could potentially help clinicians not only with diagnosis but also in objectively evaluating the severity of this disease, although further large-scale studies should first be performed to collaborate our findings.

## Figures and Tables

**Figure 1 jcm-09-00929-f001:**
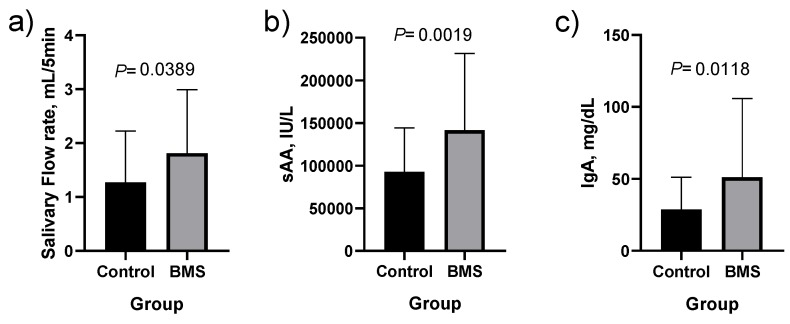
Salivary flow rate (**a**), salivary alpha-amylase (sAA; (**b**)) and immunoglobulin A (IgA; (**c**)) in healthy controls (*n* = 31) and patients with burning mouth syndrome (BMS; *n* = 51). Differences of medians and means±SD for salivary flow are 0.54 and 0.75 ± 0.41, for sAA they are 48,660 and 109,536 ± 41,492 and for IgA they are 22.47 and 34.01 ± 12.72.

**Figure 2 jcm-09-00929-f002:**
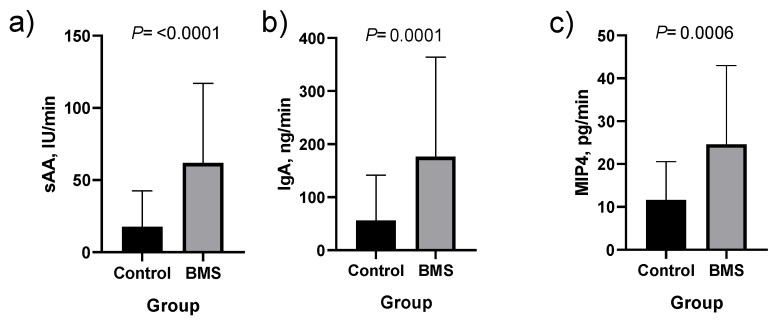
Salivary alpha-amylase (sAA; (**a**)), immunoglobulin A (IgA; (**b**)) and macrophage inflammatory protein-4 (MIP4; (**c**)) corrected by salivary flow in healthy controls (*n* = 31) and patients with burning mouth syndrome (BMS; *n* = 51). Difference of medians and means ± SD for sAA are 44.0 and 47.4 ± 15.91, for IgA are 119.9 and 147.6 ± 38.4 and for MIP4 are 13.0 and 33.6 ± 27.2.

**Figure 3 jcm-09-00929-f003:**
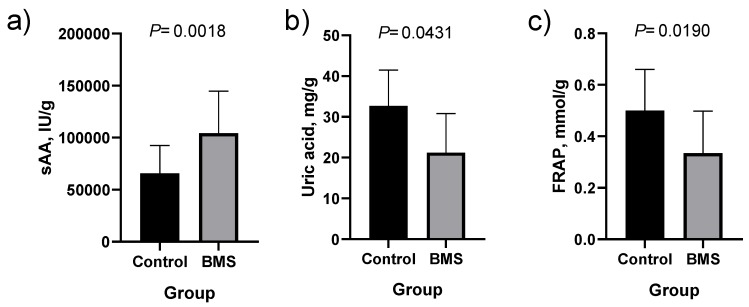
Salivary alpha amylase [sAA; (**a**)], uric acid (**b**) and ferric reducing activity [FRAP; (**c**)] corrected by salivary total protein content in healthy controls (*n* = 31) and patients with burning mouth syndrome (BMS; *n* = 51). Difference of medians and means ± SD for sAA are 38,525 and 33,653 ± 15,365, for uric acid are –11.5 and –7585 ± 4593 and for FRAP are –0.17 and –0.1413 ± 0.06614.

**Table 1 jcm-09-00929-t001:** General characteristics of participants.

Variable	Controls	SBA	*p*
Sex, *n* (%)			0.083 ^c^
Women	26 (76.5)	45 (88.2)	
Men	5 (14.7)	6 (11.8)	
Age, mean (SD) years	58.3 (11.05)	59.12 (12.29)	0.765 ^a^
Smoking, *n* (%)			0.292 ^c^
Yes	10 (32.3)	11 (21.6)	
No	18 (58.1)	29 (56.9)	
Former smoker	3 (9.7)	11 (21.6)	
Alcohol consumption, *n* (%)			0.212 ^c^
Less than once a week	21 (61.8)	31 (60.8)	
Daily	2 (5.9)	9 (17.6)	
Weekends only	11 (32.4)	11 (21.6)	
Tooth brushing, *n* (%)			0.236 ^c^
No	0 (0)	1 (2)	
Less than once a day	1 (3.2)	1 (2)	
Once a day	7 (22.6)	5 (9.8)	
Twice a day	14 (45.2)	18 (35.3)	
≥3 times a day	9 (29)	26 (51)	
Dental floss, *n* (%)			**0.006 ^c^**
No	23 (74.2)	22 (43.1)	
Yes	8 (25.8)	29 (56.9)	
Mouthwashes, *n* (%)			0.080 ^c^
No	28 (90.3)	38 (74.5)	
Yes	3 (9.7)	13 (25.5)	
Missing teeth, mean (SD)	2.1 (2.82)	5.8 (8.3)	**0.004 ^b^**
Missing teeth groups, *n* (%)			0.098 ^c^
1–5	29 (93.5)	39 (76.5)	
6–9	0 (0)	5 (9.8)	
<10	2 (6.5)	7 (13.7)	
Caries, mean (SD)	0.42 (0.99)	0.2 (1.13)	0.121 ^b^
OHIP14, mean (SD)	14.33 (1.37)	33.1 (9.02)	**<0.001 ^b^**
HAD-A, mean (SD)	3.77 (2.3)	8.71 (3.82)	**<0.001 ^b^**
HAD-D, mean (SD)	2.9 (2.4)	4.1 (3.6)	0.292 ^b^
VAS, mean (SD)	0 (0)	8.12 (1.77)	**<0.001 ^b^**
VAS groups, *n* (%)			
Mild		1 (2.0)	
Moderate		16 (31.4)	
Severe		34 (66.7)	

a, Student’s t-test; b, Mann–Whitney test; c, Pearson’s chi-square. Bold type denotes statistical significance.

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
