# Peer review of "Salivary Biomarkers and Their Correlation with Pain and Stress in Patients with Burning Mouth Syndrome"

_jcm, 2020, doi:10.3390/jcm9040929_

Round 1

Reviewer 1 Report

The authors analyzed the relationship among various salivary biomarkers and the burning mouth syndrome. The topic is important and this study was well conducted. The manuscript looks data descriptive. Before the publication, the authors should revise the manuscript to describe the data more concisely.

Since the reproducibility of quantified values is not presented, the reliability of the data cannot be evaluated. As described in the limitation paragraph, all samples were collected at a single point. Why a sample was not measured by multiple times and provide the standard deviation of each measurement? The author utilized the Mann-Whitney tests for evaluating the quantified values in two groups. Such non-parametric test does not evaluate the difference of the averaged values of the two groups and just evaluate the rank of each sample. This indicates such test will yield small P-values even in the small difference of the averaged values between two groups. Even in the data showing the significant difference using non-parametric tests, the distance of the averaged values between the two groups should be evaluated for eliminating optimistic results.

Does IgA used in study is s-IgA? The concentrations of IgA and s-IgA are different in saliva. These should be clearly defined.

Considering multiple independent tests, alpha-error should be considered and P-values should be corrected by Bonferroni or false discovery rate.

Tables 2 – 7 should be moved to Supplementary Information and instead figures showing only the important figures, e.g. the one showing significant differences should be depicted. The significant digits of the numeric values of these tables are not unified. Correct them.

At table 1, the asterisk was used as an annotation to describe the type of statistical analyses, while asterisks are usually used for the significance with p-values. The different mark should be used not for leading misunderstandings.

Title of the first “Table 4” should be changed to “Table 3”.

Shorter expressions should be used for various sentences. For example, limitation paragraph at page 9, “by means of”, “In addition” and “In order to” should be changed to “using”, “moreover”, and “to”. Such redundant expressions are observed anywhere in the manuscript.

Author Response

Dear Editor,

Please find below our responses to the queries raised by the reviewers. We have also modified accordingly the manuscript and the changes were highlighted using "Track Changes" function in Microsoft Word.

Best regards,

On behalf of all authors, Pia Lopez- Jornet

Reviewer 1.

The authors analyzed the relationship among various salivary biomarkers and the burning mouth syndrome. The topic is important and this study was well conducted. The manuscript looks data descriptive. Before the publication, the authors should revise the manuscript to describe the data more concisely.

Thanks for these nice comments.

Since the reproducibility of quantified values is not presented, the reliability of the data cannot be evaluated. As described in the limitation paragraph, all samples were collected at a single point. Why a sample was not measured by multiple times and provide the standard deviation of each measurement?

Author response: The comment of the reviewer is very interesting and it is indicated as a limitation of the manuscript. Ideally samples should have been obtained in various days, but we could not do it due to ethical reasons since patients diagnosed with BMS are prescribed with treatment and, therefore, delay of the treatment with the aim of repeated sample collection would not be the ethically appropriate.

 We would like to stress that the methods used are reliable since were precise and accurate and there are evidences that our results as reliable since confirms previous findings as in the case of IgA and sAA.

The author utilized the Mann-Whitney tests for evaluating the quantified values in two groups. Such non-parametric test does not evaluate the difference of the averaged values of the two groups and just evaluate the rank of each sample. This indicates such test will yield small P-values even in the small difference of the averaged values between two groups. Even in the data showing the significant difference using non-parametric tests, the distance of the averaged values between the two groups should be evaluated for eliminating optimistic results.

Author response: As suggested by the reviewer, we have included the differences between the medians and means of the markers that showed statistically significant differences between the two studied groups in the legends of the figures.

Does IgA used in study is s-IgA? The concentrations of IgA and s-IgA are different in saliva. These should be clearly defined.

Author response: In present study total IgA was determined. This fact was now highlighted over the manuscript.

Considering multiple independent tests, alpha-error should be considered and P-values should be corrected by Bonferroni or false discovery rate.

Author response: To the best of authors knowledge, these type of false discovery rate corrections are performed when -omic data, gen expression data, etc… are treated, since in these cases huge data sets are available and false positives have to be explored. But usually are not used when adequately validated specific methods are used, which were shown to have high accuracy and precision, unless ANOVA tests are performed comparing multiple groups (then Bonferroni corrections are performed). However, in present study, Mann-Whitney test was performed, since in our study only 2 groups were compared one to another.

Tables 2 – 7 should be moved to Supplementary Information and instead figures showing only the important figures, e.g. the one showing significant differences should be depicted. The significant digits of the numeric values of these tables are not unified. Correct them.

Author response: As suggested by the reviewer we have prepared 3 figures and moved the tables 2-7 to the supplementary material. The tables were also revised in order unify the numeric digits.

At table 1, the asterisk was used as an annotation to describe the type of statistical analyses, while asterisks are usually used for the significance with p-values. The different mark should be used not for leading misunderstandings.

Author response: The asterisk was changed by the letter “c”

Title of the first “Table 4” should be changed to “Table 3”.

Author response: The table number was modified accordingly. Sorry for this mistake.

Shorter expressions should be used for various sentences. For example, limitation paragraph at page 9, “by means of”, “In addition” and “In order to” should be changed to “using”, “moreover”, and “to”. Such redundant expressions are observed anywhere in the manuscript.

Author response: the manuscript was revised and modified accordingly in order to reduce redundant expressions.

Reviewer 2 Report

See attached. 

Author Response

This manuscript seems to be well written. I do have some comments, however.

Thanks for these nice comments.

  1. It is not stated in the manuscript the importance of linking the studied biomarkers to the condition of BMS. What are the implication/significance of your findings?

Author response: In order to stress this, we have included in the Page 7 line 283 the following: Overall, tacking in account the complicated diagnosis of BMS, biomarkers that were shown to change differently between groups could potentially help clinicians not only in the diagnosis but also in the objective severity evaluation of this disease, although first further large-scale studies should be performed to collaborate our findings. If the reviewer thinks that we should include more information we would be pleased to do it.

  1. It is not mentioned what is the statistical analysis way used to test correlation. I would also encourage the journal to consult a statistician.

Author response: Maybe there was some misunderstanding, since in the statistical analysis section and in the legends of the tables 5-7 it was indicated that the correlations between variables were checked using the partial correlation corrected by age and sex.

  1. Please state how you have determined the sample size used.

Author response: Previous studies that also analyse biomarkers in saliva in patients with burning mouth syndrome, such as SAA and total IgA, have been able to detect differences with equal or lower than 30 individuals in each group considering α = 0.05 and β= 0.20 (e.g. Imura et al., 2016). Based in these data we assumed that we had enough power (ncontrols=31; nBMS=51) to achieve our aims. We have included this information in the new version of the manuscript,2.5. Statistical analysis section.

Imura, H.; Shimada, M.; Yamazaki, Y.; Sugimoto, K. Characteristic changes of saliva and taste in burning mouth syndrome patients. J. Oral Pathol. Med. 2016, 45, 231–6.

  1. Please explain more about the scales used to measure variables, and how they were applied.

Author response: The scales used were applied as indicated by the previously reported studies, which validate the use of these variables and of which references are given in the text. All the patients were evaluated by a trained professional (CCF) who conducted the interviews and explorations, administered the questionnaires and collected the saliva samples.

  1. Tables 4 and 4 (should be 3 and 4) need better, more informative titles.

Author response: Sorry for this mistake. Nevertheless, according to the reviewer 1 indications, tables 2-4 were integrated into one and passed to Supplementary material 1.

  1. I am not sure about the importance of findings presented in Tables 5, 7, and 7. Please clarify.

Author response: These tables were passed to Supplementary material 2 and only statistically significant correlations between salivary biomarkers and clinical variables were left in the text of the result section.

Reviewer 3 Report

This is an observational study worked on possible biomarkers in burning mouth syndrome (BMS). Authors have investigated 11 biochemical substances and reported some interesting results. The topic is interesting and the null hypothesis is acceptable, although the study design has some critical problems.

Major concerns:

There are at least two critical and essential problems in this study.

  1. Authors have not excluded some major diseases that affect salivary flow rate, e.g. Sjögren’s syndrome and thyroid diseases in diagnosing BMS. This may have influenced the conflicting result in resting salivary flow. Authors are also recommended to measure not only resting salivary flow but also stimulated salivary flow to observe the disease characteristics. Biochemical measurement can be done with collected resting saliva after this observation. Many previous studies have reported that BMS patients likely show decreased resting flow but not stimulated salivary flow. This may be due to damage of minor salivary glands (Imamura et al. Journal of Oral Rehabilitation 46, 2019). Patients with Sjögren’s syndrome and thyroid diseases may show reduction both in resting and stimulated salivary flow.
  2. Authors have not investigated the influence of occult blood in saliva. Occult blood from periodontal tissues seriously affects measurements in proteins and biomarkers (Kang et al. Clinical Chemistry and Laboratory Medicine 57, 2018). This problem cannot be neglected. All the studies using saliva should be aware of this problem.

Minor concerns:

  1. There are many abbreviations that are not spelled out in the text.
  2. The text has some typographical errors.

Author Response

Thanks for these nice comments.

Reviewer 3.

This is an observational study worked on possible biomarkers in burning mouth syndrome (BMS). Authors have investigated 11 biochemical substances and reported some interesting results. The topic is interesting and the null hypothesis is acceptable, although the study design has some critical problems.

Major concerns:

There are at least two critical and essential problems in this study.

  1. Authors have not excluded some major diseases that affect salivary flow rate, e.g. Sjögren’s syndrome and thyroid diseases in diagnosing BMS. This may have influenced the conflicting result in resting salivary flow. Authors are also recommended to measure not only resting salivary flow but also stimulated salivary flow to observe the disease characteristics. Biochemical measurement can be done with collected resting saliva after this observation. Many previous studies have reported that BMS patients likely show decreased resting flow but not stimulated salivary flow. This may be due to damage of minor salivary glands (Imamura et al. Journal of Oral Rehabilitation 46, 2019). Patients with Sjögren’s syndrome and thyroid diseases may show reduction both in resting and stimulated salivary flow.

Author response: the patients diagnosed with Sjögren’s syndrome, thyroid diseases or other pathologies other than BMS were not included in the study. In order to avoid misunderstanding we have highlighted this fact in the M&M, which can now be read as follows (Page 2, line 63): “Cancer patients and individuals with known liver or kidney disease were excluded, as were those with other than BMS oral diseases such as Sjögren syndrome or ongoing infection, thyroid diseases, coagulation disorders, psychiatric disease, and pregnant or nursing women.

  1. Authors have not investigated the influence of occult blood in saliva. Occult blood from periodontal tissues seriously affects measurements in proteins and biomarkers (Kang et al. Clinical Chemistry and Laboratory Medicine 57, 2018). This problem cannot be neglected. All the studies using saliva should be aware of this problem.

Author response:  We did evaluate for the presence of blood in saliva visually. In order to clarify this, in the current form of the manuscript, the following sentence was included: “Only samples free from haemolysis determined by visual inspection were used in the study [19,20].” In addition, among the limitations of the study, the following sentence was included: “Finally, the possible blood contamination in saliva samples should have been evaluated using more sensitive methods, such as determination of hemoglobin or transferrin, although these methods were shown to be affected by different factors such as age, hormones, salivary flow among others [19].”

Minor concerns:

  1. There are many abbreviations that are not spelled out in the text.

Author response: We revised the manuscript accordingly and spelled missing abbreviations.

  1. The text has some typographical errors.

Author response: we have done our best in revising the typographical errors in the text.

Reviewer 4 Report

Thank you to the authors for an interesting and thought provoking article. The article is clear and with an acceptable standard of English requiring only a few minor changes. 

The study looks at the concentration of biomarkers in the saliva of BMS patients versus controls.

1. Although the IASP BMS definition is stated in the introduction, the diagnostic criteria used for identification of BMS is not expressly stated in the methods. This should be done in accordance with STROBE guidelines.

2. No reasoning for sample size is given. Again this should be done in accordance with STROBE guidelines.

In resting whole saliva significant differences in the concentrations of sIgA and sAA were seen between BMS patients and controls. These are designated as markers of innate immune and adrenergic systems respectively. These results are not new but do confirm previous findings.

Not until the samples are correct for salivary flow and total protein are differences seen in MIP-4 (a marker of inflammation) and FRAP and uric acid, markers of oxidative stress.

3. While these subsequent results are interesting, there is very little rationale given for correcting for salivary flow or total protein.

I think it would benefit the manuscript if this was explained in the text with a clear rationale. 

4. Some minor grammatical and spelling issues have been identified on the annotated PDF attached. 

Author Response

Thank you to the authors for an interesting and thought provoking article. The article is clear and with an acceptable standard of English requiring only a few minor changes. 

Thanks for your nice words.

The study looks at the concentration of biomarkers in the saliva of BMS patients versus controls.

  1. Although the IASP BMS definition is stated in the introduction, the diagnostic criteria used for identification of BMS is not expressly stated in the methods. This should be done in accordance with STROBE guidelines.

Author response: The diagnostic criteria were added into the M&M section as suggested by reviewer. In addition, we revised STORBE guidelines and made changes in order to complete missing information in the manuscript.

  1. No reasoning for sample size is given. Again this should be done in accordance with STROBE guidelines.

Author response: please see the response to the Reviewer 2 comment 3 about sample size.

  1. In resting whole saliva significant differences in the concentrations of sIgA and sAA were seen between BMS patients and controls. These are designated as markers of innate immune and adrenergic systems respectively. These results are not new but do confirm previous findings.

Not until the samples are correct for salivary flow and total protein are differences seen in MIP-4 (a marker of inflammation) and FRAP and uric acid, markers of oxidative stress.

 While these subsequent results are interesting, there is very little rationale given for correcting for salivary flow or total protein. I think it would benefit the manuscript if this was explained in the text with a clear rationale.

Author response: In accordance to reviewers indications the following text was added in the manuscript in order to add rationale for correcting for salivary flow and total protein content: “Since components present in saliva have different origins - are locally secreted by salivary glands or oral mucosa, or pass from blood among other, and can be affected by not uniform salivary flow rate, the need of salivary biomarker values correction by flow rate, total protein content, or salivary osmolarity vs. use of absolute their value is being increasingly acknowledged and studied by different authors [25,28].” If the reviewer thinks that more information should be provided we would be happy to do it.

  1. Some minor grammatical and spelling issues have been identified on the annotated PDF attached.

Author response: the indicated grammatical and spelling issues were incorporated in the current form of the manuscript. Thanks a lot.

Round 2

Reviewer 2 Report

In the third page please let the reader know how many samples with hemolysis, if any, were discarded. 

Author Response

Reviewer 2

thanks for your comments

In the third page please let the reader know how many samples with hemolysis, if any, were discarded.

Author response. No samples were discarded in the patients used in the study because criteria for being included in the study was that sample would not present hemolysis during visual inspection. In order to clarify this the following sentence was included in Page 2, Line 71: “patients were not included in the study if their samples presented hemolysis as determined by visual inspection [19,20].”

Reviewer 3 Report

I cannot ensure the quality of data obtained in this study. All the explanations to my queries are added as an afterthought. Authors described "Only samples free from hemolysis as determined by visual inspection were used in the study." This makes no sense. (Saliva proteome research: current status and future outlook. Schulz et al. Critical Reviews in Biotechnology, 2013; 33(3): 246–259) 

Author Response

thanks for your comments

Reviewer 3

I cannot ensure the quality of data obtained in this study. All the explanations to my queries are added as an afterthought. Authors described "Only samples free from hemolysis as determined by visual inspection were used in the study." This makes no sense. (Saliva proteome research: current status and future outlook. Schulz et al. Critical Reviews in Biotechnology, 2013; 33(3): 246–259)

Author response. Sorry if this sentence was not clear. With this sentence we wanted to indicate that criteria for the samples to be used in the study was that they should not have haemolysis by visual inspection. We have clarified this and now in the manuscript it can be read ( Page 2, Line 71) – “In all cases, patients were not included in the study if their samples presented hemolysis as determined by visual inspection [19,20].”

And we are sorry that the reviewer thinks that this explanation about the control for haemolysis was added as an afterthought. Regarding this we wanted to stress that evaluation of presence of hemolysis in saliva samples is routine procedure in our laboratory and therefore is not an afterthought. This can be seen from our previous published studies, e.g.:

  • doi: 10.1007/s00784-020-03214-7: “A total of 218 participants meeting the inclusion and exclusion criteria were included in the study. However, 66 samples were excluded because of low volume (n = 47); contamination with food traces, medicines, lipstick, etc. (n= 16); labeling error (n =2); or visible hemolysis (n = 1)”.
  • doi: 10.1155/2018/5187549: “Samples with blood contamination (determined by visual inspection scale) were excluded”
  • doi: 10.1111/jop.12522: “Samples with blood contamination (determined by visual inspection) were excluded”